# RNA-Seq Insight into the Impact and Mechanisms of Methyl Donor and Glycine Betaine Osmoprotectant on Polyketide Secondary Metabolism in *Monascus purpureus* M1

**DOI:** 10.3390/jof11040273

**Published:** 2025-04-01

**Authors:** Zheng Liu, Haijing Zhang, Furong Xue, Lidan Niu, Chenchen Qi, Wei Chen, Jie Zheng, Chengtao Wang

**Affiliations:** 1Key Laboratory of Geriatric Nutrition and Health, Ministry of Education, Beijing Engineering and Technology Research Center of Food Additives, School of Food and Health, Beijing Technology & Business University (BTBU), Beijing 100048, China; lz1622176788@163.com (Z.L.); haijingvv@yeah.net (H.Z.); hf15375088475@163.com (F.X.); weichen@btbu.edu.cn (W.C.); 2School of Chemistry and Chemical Engineering, Chongqing University, Chongqing 401331, China; niulidan@cqifdc.org.cn; 3Key Laboratory of Condiment Supervision Technology for State Market Regulation, Chongqing Institute for Food and Drug Control, Chongqing 401121, China; 4Xinjiang Xinkang Agricultural Development Co., Ltd., Urumqi 830032, China; qichenchenxj@163.com

**Keywords:** *Monascus purpureus*, glycine betaine, *Monascus* pigments, monacolin K, RNA-seq

## Abstract

Glycine betaine (GB) serves as both a methyl donor and osmoprotectant in microorganisms, facilitating growth and enhancing metabolic product yields. While the polyketide metabolites from *Monascus purpureus*, such as *Monascus* pigments (MPs) and monacolin K (MK), have been extensively studied, the effects of GB on their production and the underlying molecular mechanisms remain insufficiently explored. In this study, various concentrations of GB were added to *Monascus purpureus* M1 cultures, followed by RNA sequencing, RT-qPCR, differential gene expression analysis, and functional enrichment to investigate the regulatory impact of GB on polyketide metabolism. Protein–protein interaction network analysis identified key upregulated genes, including RPS15, RPS14, RPS5, NDK1, EGD2, and ATP9, particularly during the later growth phases. GB significantly upregulated genes involved in stress response, secondary metabolism, and polyketide biosynthesis. Scanning electron microscopy, HPLC, and UV-Vis spectrophotometry further confirmed that GB promoted both strain growth and polyketide production, with red pigment and MK production increasing by 120.08% and 93.4%, respectively. These results indicate that GB enhances growth and polyketide metabolism in *Monascus purpureus* by functioning as both a methyl donor and osmoprotectant, offering new insights into optimizing microbial polyketide production and revealing gene regulatory mechanisms by GB in *Monascus purpureus*.

## 1. Introduction

*Monascus purpureus* is a filamentous fungus belonging to the *Ascomycota* phylum [1]. This microorganism is renowned for its ability to produce a variety of secondary metabolites with significant health benefits, including *Monascus* pigments (MPs), monacolin K (MK), and *γ*-aminobutyric acid (GABA) [2]. Consequently, *M. purpureus* has been extensively studied and applied in fields such as pharmacology [2,3] and the food industry [4,5]. The growing interest in *Monascus* is driven by its promising production capabilities and the expanding range of applications, signaling the importance of ongoing research. As demand for *M. purpureus* products continues to rise, this fungus is expected to present numerous development opportunities, underscoring the need for intensified research efforts.

Monacolin K (MK), also known as lovastatin, is one of the key bioactive compounds produced by *M. purpureus*, with multiple health benefits. Research by A. F. G. Cicero [6] has highlighted MK’s potent cholesterol-lowering effects, positioning it as a core active ingredient in cholesterol-lowering functional foods [7]. Furthermore, L. Zhang employed weighted gene co-expression network analysis (WGCNA) and Connectivity Map (CMap) to identify MK’s potential anti-gastric cancer properties [8]. Additional studies by C. C. Chen et al. demonstrated that MK can induce apoptosis in HL-60 cancer cells, thereby inhibiting acute myeloid leukemia (AML). Moreover, P. Divsalar [9] found that MK exhibits potential antidepressant effects. As a result, enhancing the production of MK during the fermentation of *M. purpureus* has become a major focus of research. For instance, studies by Zhang et al. demonstrated that the addition of glutamic acid [10] and arginine [11] to the fermentation media significantly increases MK yield.

*M. purpureus* is also capable of synthesizing three types of *Monascus* pigments (MPs)— yellow, orange, and red—each of which offers various health benefits. These pigments have been the subject of extensive research. For example, H. Tan et al. [12] reported that the yellow pigment possesses antitumor and antioxidant activities, while C. Z. Zhao et al. [13] found that MPs can treat non-alcoholic fatty liver disease by modulating lipid metabolism. Additionally, H. P. Mohankumari et al. [14] reported that MPs have cholesterol-lowering effects, contributing to improved cardiovascular health. In the food industry, MPs are valued not only for their health benefits but also for their ability to enhance flavor and texture. H. K. Abdelhakim et al. [15] demonstrated that MPs can inhibit microbial growth, thereby extending the shelf life of food products. Moreover, MPs provide vibrant color to foods, such as red fermented bean curd, which can stimulate appetite [4].

Numerous studies have shown that the addition of exogenous inducers can modulate the growth of *M. purpureus* and the synthesis of its secondary metabolites. For instance, the addition of the surfactant Triton X-100 to the fermentation medium has been shown to increase polyketide metabolite yields, and it was found that adding linoleic acid significantly enhances the yield of polyketide metabolites [16,17]. Similarly, Shi et al. [18] discovered that the quorum-sensing molecule (QSM) *γ*-butyrolactone positively regulates polyketide metabolite production during fermentation. Additionally, Yin et al. [19] reported that the inclusion of various amino acids in the fermentation medium can positively influence polyketide metabolite production. In our study, we explored the effects of adding glycine betaine (GB) during the fermentation of *M. purpureus* on the yield of polyketide metabolites.

Glycine betaine (GB) is an alkaloid commonly used as a methyl donor. It plays a crucial role in methyl metabolism across various organisms, including *Pseudomonas aeruginosa*, and *Agrobacterium tumafaciens* [20], vertebrates [21,22], *Beta vulgaris* ssp., *Bougainvillea* spp., and others [23]. Therefore, GB has a significant impact on microbial growth and metabolism. Studies have shown that adding GB to the culture medium not only accelerates cell growth, but also increases the yield of both primary [24,25] and secondary metabolites [26,27]. GB is also a common osmoprotectant. In many microorganisms, it acts as an osmolyte to counteract environmental stresses, such as low temperatures, drought, and high salinity [20,28]. Many microbes either uptake or synthesize GB to withstand these stresses. For instance, Y. Nakagawa et al. [29] found that GB provides protective effects during enzyme freeze denaturation processes, while X. Hu et al. [30] reported that GB enhances the osmotolerance of microbial cells, thereby improving salt tolerance. Moreover, Wang et al. [31] discovered that GB mitigates the negative impacts of high salt stress on gut microbiota in mice. Tang et al. [32] observed that GB aids *Pseudomonas protegens* SN15-2 in adapting to hyperosmotic stress. Additionally, some archaea can spontaneously synthesize GB to combat osmotic stress [33].

Numerous studies have investigated the effects of various compounds on polyketide production in *M. purpureus*; yet, there has been limited focus on the underlying genetic regulatory mechanisms. Additionally, the impact of methyl donors on polyketide biosynthesis in *Monascus* has not been previously examined. This study addresses this gap by exploring the dual role of GB as both a methyl donor and an osmoprotectant in enhancing polyketide biosynthesis. Using RNA-seq technology, we investigated the transcriptomic landscape of *M. purpureus* M1 under GB treatment at various time points. Through differential gene expression analysis, protein–protein interaction (PPI) network mapping, and real-time quantitative polymerase chain reaction (RT-qPCR) validation, we examined the genetic regulatory mechanisms by which GB influences the biosynthesis of MK and MPs. In addition, high-performance liquid chromatography (HPLC) and UV-Vis spectrophotometry were employed to quantify MK and MPs production, while scanning electron microscopy (SEM) provided insights into morphological changes in spores, mycelium, and extracellular secretions. The findings revealed that GB activates key pathways involved in polyketide metabolism, leading to the upregulation of genes linked to secondary polyketide production and stress resistance. This study reveal into the regulatory mechanisms through which GB enhances polyketide metabolite synthesis, highlighting the intricate relationship between methyl donors and microbial stress response pathways. Additionally, it offers a novel perspective on utilizing exogenous compounds to boost the yield of polyketide secondary metabolites.

## 2. Materials and Methods

### 2.1. Materials

Methanol (analytical grade) and ethanol (analytical grade) were purchased from Beijing Myrida Technology Co., Ltd. (Beijing, China), and Beijing Boao Tuoda Technology Co., Ltd. (Beijing, China). Anhydrous GB (purity ≥ 99%) was supplied by Beijing Boao Tuoda Technology Co., Ltd. (Beijing, China). Potato dextrose agar (PDA), glucose, peptone, soybean powder, glycerol, KH_2_PO_4_, NaNO_3_, MgSO_4_·7H_2_O, and ZnSO_4_·7H_2_O were obtained from National Pharmaceutical Group Chemical Reagent Co., Ltd. (Beijing, China) The FastQuant cDNA Synthesis Kit, SuperReal PreMix Plus (SYBR Green) Kit, RNAprep Pure Polysaccharide Polyphenol Plant Total RNA Extraction Kit were procured from TianGen Biochemical Technology (Beijing) Co., Ltd. (Beijing, China).

### 2.2. Medium

Strain activation medium: composed of 3.7% potato dextrose agar (PDA), with a natural pH.

Liquid seed medium [18]: formulated with glucose (30 g/L), soybean meal (15 g/L), KH_2_PO_4_ (2 g/L), NaNO_3_ (2 g/L), MgSO_4_·7H_2_O (1 g/L), glycerol (70 g/L), and peptone (10 g/L). Sterilized at 115 °C for 20 min under high temperature and pressure.

Liquid fermentation medium [34]: contained rice powder (20 g/L), glycerol (90 g/L), peptone (10 g/L), KH_2_PO_4_ (2.5 g/L), NaNO_3_ (5 g/L), MgSO_4_·7H_2_O (1 g/L), and ZnSO_4_·7H_2_O (2 g/L). Sterilized at 121 °C for 15 min under high temperature and pressure.

### 2.3. STRAIN Activation and Fermentation

The *M. purpureus* M1 strain, preserved in our laboratory, was activated by subculturing for two generations on PDA medium to prepare a spore suspension. The prepared spore suspension was added to 50 mL of seed medium at a 5% (*v*/*v*) inoculum volume and incubated at 30 °C with shaking at 200 rpm for 48 h [35]. GB solutions with volume fractions of 0%, 0.01%, 0.1%, 1%, and 10% (stock solution concentration: 1 mol/L) were added to 50 mL of *Monascus* liquid fermentation medium. The seed culture was inoculated into 50 mL of fermentation medium at a 10% (*v*/*v*) inoculation volume. Fermentation was initially conducted at 30 °C with shaking at 200 rpm for 48 h, followed by a temperature shift to 25 °C and 150 rpm, and cultivation was continued for 15 days [36,37]. Changes in secondary metabolite production were monitored throughout the fermentation process.

### 2.4. Analysis of Pigment Production

The *Monascus* fermentation broth was mixed with 70% ethanol in a 1:3 volume ratio and incubated in the dark at 60 °C in a water bath for 1 h. The mixture was then centrifuged at 5000 rpm for 10 min, and the supernatant was collected. The absorbance of the supernatant was measured using a UV-Vis spectrophotometer at 410 nm, 448 nm, and 505 nm. The color values of the yellow pigment (410 nm), orange pigment (448 nm), and red pigment (505 nm) were calculated using the following formula: color value = absorbance × dilution factor [34,38]*Monascus* yellow pigment color value (U/mL) = OD_410_ × dilution factor*Monascus* orange pigment color value (U/mL) = OD_448_ × dilution factor*Monascus* red pigment color value (U/mL) = OD_505_ × dilution factor

### 2.5. HPLC Analysis of MK Production

Fermentation broth samples were collected on days 2, 5, 8, 12, and 15, and mixed with 75% methanol in a 1:3 volume ratio, followed by ultrasonic extraction for 30 min (250 W, 40 kHz). After standing in the dark for 30 min, the mixture was filtered through a 0.22 µm organic membrane filter. The MK yield was quantified using HPLC (LC-20AT, Shimadzu, Kyoto, Japan) [39].

Chromatographic Conditions: HPLC analysis was performed on an Inertsil ODS-3 C_18_ column (150 mm × 4.6 mm × 5 µm). The mobile phase consisted of 0.1% H_3_PO_4_ and methanol in a 1:3 (*v*/*v*) ratio, with a flow rate of 1 mL/min and a column temperature of 30 °C. Detection was conducted using an ultraviolet detector at 237 nm, with an injection volume of 10 µL [36].

### 2.6. SEM Analysis of *Monascus* Mycelium and Spores

The effects of GB on the growth and development of *Monascus* mycelium and spores were examined using a scanning electron microscope (Hitachi Su8020, Hitachi, Tokyo, Japan) [35]. Mycelium samples (1 cm^3^) were subjected to low-speed centrifugation at 2000 g to remove the supernatant. The samples were fixed overnight in 2.5% glutaraldehyde, then washed twice with PBS for 10 min each time, followed by fixation in 1% osmium tetroxide pre-cooled to 4 °C for 1 h. After washing twice with PBS for 10 min each time, the samples were dehydrated in a graded series of ethanol concentrations (50%, 70%, 100%, and 100%), each for 15 min and repeated twice. Subsequently, the samples were transferred from 100% ethanol to a critical point dryer (Tousimis Autosamdri-815, Series A, Rockville, MD, USA), where ethanol was replaced with liquid carbon dioxide for 40 min. The temperature and pressure were then increased to the critical point of carbon dioxide and maintained for 4 min, after which the carbon dioxide was slowly released over 30 min before sample removal. The samples were gold-coated using a vacuum sputter coater (Hitachi MC1000) at 10 kV for 220 s and examined under the scanning electron microscope.

### 2.7. RT-qPCR Analysis of MPs and MK Biosynthesis-Related Genes

Relative gene expression analysis was performed using RT-qPCR as described by Zhang [40]. Total RNA was extracted from 1 mmol/L GB treatment and control groups *Monascus* mycelium using the Polysaccharide Polyphenol Total RNA Extraction Kit (TianGen, Beijing, China). First-strand cDNA was synthesized using the FastQuant RT Kit (with gDNase) (TianGen, Beijing, China) and the SuperReal PreMix (SYBR Green) (TianGen, Beijing, China). RT-qPCR was carried out on a CFX96 Real-Time PCR Detection System (Bio-Rad, Hercules, CA, USA). The amplification protocol consisted of an initial denaturation at 95 °C for 15 min, followed by a three-step PCR cycle (denaturation at 95 °C for 10 s, annealing at 60 °C for 20 s, and extension at 72 °C for 30 s) repeated for 45 cycles [36,40]. Relative gene expression levels were calculated using the 2*^−^*^∆∆*Ct*^ method, normalized to the transcription level of the GAPDH gene. Primer sequences were designed using Primer Premier 5 software, and each sample was tested in triplicate by RT-qPCR. Primers for MPs biosynthesis genes included those for mokA-mokI (GenBank accession number: DQ176595.1), MPs biosynthesis genes (GenBank accession number: MK764693.1). The spore development-related genes *wetA*, *brlA*, the global regulator *laeA*, and GAPDH (GenBank accession number: HQ123044.1) are shown in Appendix A.

### 2.8. RNA-Seq Data Analysis

RNA sequencing and transcriptomic data analysis were conducted following established protocols. Sequencing was carried out by General Biol (Anhui, China) Co., Ltd. A total of 1 μg of RNA per sample was used for library preparation. Poly(A) mRNA was isolated using oligo(dT) beads, and mRNA fragmentation was achieved through the use of divalent cations and elevated temperatures. cDNA synthesis was carried out using random primers for first-strand synthesis, followed by second-strand synthesis. The double-stranded cDNA was purified and subjected to end repair and dA-tailing in a single reaction, after which T-A ligation was performed to add adaptors to both ends. Adaptor-ligated DNA was size-selected using DNA Clean Beads, and each sample was amplified by PCR using P5 and P7 primers. PCR products were validated, and the libraries, indexed for multiplexing, were sequenced on an Illumina HiSeq, Illumina Novaseq (Illumina, Inc. San Diego, CA, USA), or MGI2000 instrument (MGI Tech Co., Ltd., Shenzhen, China) with a 2 × 150 paired-end (PE) configuration, following the manufacturer’s instructions.

For functional analysis, GOSeq (v1.34.1) was used to identify enriched gene ontology (GO) terms for the differentially expressed genes (DEGs), with a significance threshold of padj ≤ 0.05. KEGG pathway enrichment was performed using in-house scripts to identify significant DEGs, with results visualized using ChiPlot (https://www.chiplot.online/) accessed on 2 November 2024. Differential expression analysis was conducted using the DESeq2 Bioconductor package Version 1.26.0, which models data based on a negative binomial distribution. Dispersion and logarithmic fold change estimates were improved through data-driven prior distributions. Multiple test correction was applied using the Benjamini/Hochberg (BH) method, and a *p*-value < 0.05 was considered statistically significant at the 95% confidence level.

### 2.9. PPI Network Construction and the Identification of Hub Genes

The construction of the protein–protein interaction (PPI) network and the identification of hub genes followed a multi-step procedure. First, interaction data were gathered and preprocessed using STRING (Version 12.0, https://www.string-db.org/) and Cytoscape (Version 3.10.2) software [41]. The PPI network was built based on experimental data and predictions from gene neighborhoods, gene fusions, co-occurrence, co-expression, and text mining, with a medium confidence score of 0.4 (Talebi et al. [42]). Key hub genes were identified through topological analyses using CytoHubba, a Cytoscape plugin [43]. Eight different metrics were used for hub gene identification: degree, maximum neighborhood component (MNC), maximal clique centrality (MCC), edge percolated component (EPC), closeness, betweenness, bottleneck, and radiality. Functional analysis of the hub genes was conducted using ClueGO and CluePedia to interpret their biological relevance in the context of the study [44].

### 2.10. Statistical Analysis

Each experiment was performed in triplicate. Statistical analyses were conducted using one-way analysis of variance (ANOVA) with GraphPad Prism 10.2.3. Numerical data are expressed as mean ± SD, and *p*-values of <0.05, <0.01, <0.001, and <0.0001 were considered statistically significant.

## 3. Results

### 3.1. Effects of GB on the Production of MPs and MK

The impact of exogenous GB supplementation on the production of MPs and MK by *M. purpureus* M1 was assessed using UV-Vis spectrophotometry and HPLC. The findings indicated that both 1 mmol/L and 100 mmol/L GB significantly enhanced the production of MPs compared to the control group. On day 8 of fermentation, the addition of 100 mmol/L GB resulted in the highest increase in MPs production, with yellow, orange and red pigments levels increasing by 88.43%, 92.03%, and 97.55%, respectively. By day 12, the 1 mmol/L GB treatment exhibited the greatest effect on MPs production, with yellow, orange, and red pigments levels rising by 112.59%, 119.54%, and 120.08%, respectively. However, by day 15, MPs production in the 1 mmol/L GB group had decreased compared to day 12, though it remained significantly higher than in the control group (Figure 1a–c).

In summary, the addition of 1 mmol/L GB led to a peak in MPs production on day 12, after which yields began to decline. Moreover, the 100 mmol/L GB treatment resulted in a peak on day 8, although its overall effect was less pronounced than that of the 1 mmol/L treatment. Additionally, both groups exhibited a marked decrease in MPs yield after reaching their respective peaks.

Similarly, the addition of 1 mmol/L and 10 mmol/L GB significantly increased MK production on day 12 (Figure 1d), with respective increases of 93.40% and 81.26%. By day 15, MK production reached its highest level of 89.74 mg/L during the 15-day fermentation process. The 10 mmol/L GB treatment resulted in a 49.85% increase in MK production compared to the control group. These results demonstrate that GB has a significant regulatory effect on *Monascus* fermentation products, enhancing the yield of secondary metabolites such as MPs and MK.

### 3.2. RT-qPCR Analysis of MK Biosynthesis-Related Genes

To further elucidate the effect of GB on the expression of genes involved in MK biosynthesis in *Monascus*, RT-qPCR was employed to measure the relative expression levels of the genes *mokA*, *mokB*, *mokC*, *mokD*, *mokE*, *mokF*, *mokG*, *mokH*, and *mokI*. The results (Figure 2) are presented in the accompanying figure. On day 5 of fermentation, the expression levels of *mokD*, *mokF*, and *mokI* were upregulated by 1.60-fold, 2.44-fold, and 3.41-fold, respectively, compared to the control group. On day 8, the expression levels of *mokA*, *mokD*, and *mokE* were upregulated by 1.95-fold, 3.92-fold, and 4.12-fold, respectively. By day 12, the expression levels of *mokD*, *mokF*, *mokG*, *mokI*, and *mokH* were upregulated by 4.40-fold, 2.88-fold, 2.34-fold, 2.73-fold, and 6.37-fold, respectively. These findings indicate that GB can significantly regulate the expression of genes involved in MK biosynthesis at various stages of fermentation, providing valuable insights into its mechanism of action.

### 3.3. RT-qPCR Analysis of MPs Biosynthesis-Related Genes

The effect of GB on the expression levels of genes involved in MPs biosynthesis was also investigated using RT-qPCR. Twelve related genes (*mppA*, *mppB*, *mppC*, *mppD*, *mppE*, *mppG*, *mpp7*, *mppR1*, *mppR2*, *mpPKS5*, *mpFasA2*, and *mpFasB2*) were analyzed, and the results are depicted in Figure 3. On day 2 of fermentation, the expression levels of *mpFasA2* were slightly upregulated, increasing 1.87-fold. On day 5, the expression levels of *mpPKS5* and *mppA* were upregulated 1.26-fold and 2.08-fold, respectively. On day 8, the expression levels of *mppC* increased by 3.45-fold, respectively. By day 12, the expression levels of *mppB*, *mppC*, *mppD*, *mppE*, *mpp7*, *mppR2*, *mpPKS5* and *mpFasB2* were significantly upregulated, with increases of 6.50-fold, 3.88-fold, 4.97-fold, 1.78-fold, 6.13-fold, 2.09-fold, 3.34-fold, and 2.94-fold, respectively. These results suggest that GB promotes the expression of genes associated with MPs biosynthesis, and that the expression levels of different genes vary significantly across different fermentation stages. These findings provide important insights for further research into the regulatory pathways of MPs biosynthesis.

### 3.4. Effects of GB on the Morphology of Monascus Spores and Mycelia

The effects of exogenously added GB on the morphology of *Monascus* mycelia and spores were observed using SEM at magnifications of 2000×, 8000×, and 10,000×. The results revealed that (Figure 4a), compared to the control group, spores and mycelia in the GB-treated group exhibited significant wrinkling and folding. Additionally, more extracellular secretions were observed around the mycelia in the GB-treated group. These morphological changes suggest that the addition of GB may alter the growth patterns of spores and mycelia, leading to enhanced secretion of extracellular substances and, consequently, increased production of MPs and MK. Additionally, RT-qPCR was performed on the global regulator (*laeA*) and two central regulators (*brlA* and *wetA*) in *M. purpureus* M1 to investigate the impact of GB on spores at the transcriptional level. As shown in Figure 4b, the results indicated that in GB treatment, the expression levels of *brlA*, *laeA*, and *wetA* genes increased by 3.54-, 2.54-, and 2.18-fold, respectively, compared to those in the control.

### 3.5. RNA-Seq Correlation Check and Principal Component Analysis

To investigate the potential molecular mechanisms by which GB enhances MPs and MK synthesis, RNA-seq was employed to study transcriptomic changes following GB treatment. Low-quality reads were removed, and adapter sequences were trimmed. The clean reads showed Q30 values above 94% and GC content exceeding 50.5%, ensuring high data quality. Reference genome sequences and gene model annotation files from UCSC, NCBI, and ENSEMBL databases were used. Hisat2 (v2.2.1) was employed to index and align the clean reads to the reference genome. RNA sequencing was performed in triplicates for both the control and GB-treated groups. Pearson’s correlation coefficient was calculated to assess the reproducibility of biological replicates, with values exceeding 0.86, demonstrating high similarity among replicates (Figure 5a). Principal component analysis (PCA) showed distinct clustering within each group, reflecting high internal similarity and significant differences between the control and GB-treated samples (Figure 5b).

### 3.6. Comparison of Differential Genes

Differentially expressed genes (DEGs) were analyzed using DESeq2, and a volcano plot was used to visualize these DEGs. On day 2, the GB-treated group showed 594 DEGs compared to the control group, with 408 upregulated and 186 downregulated (Figure 6a). On day 12, there were 932 DEGs, with 336 upregulated and 596 downregulated (Figure 6b). A total of 89 DEGs were shared between day 2 and day 12 (Figure 6c). Hierarchical clustering based on log-transformed FPKM values identified distinct gene expression patterns (Figure 6d).

### 3.7. Functional Enrichment Analysis of DEGs

Gene enrichment and functional annotation of the DEGs were performed using GOSeq and topGO. On day 2, 152 DEGs were annotated, with 14 related to biological processes, 3 to cellular components, and 13 to molecular functions (Figure 7a). On day 12, 110 DEGs were annotated, with 11 related to biological processes, 4 to cellular components, and 13 to molecular functions (Figure 7b). KEGG pathway enrichment analysis showed that on day 2, the most significantly regulated pathways were related to metabolic pathways and the biosynthesis of secondary metabolites (Figure 7c). On day 12, the ribosome and oxidative phosphorylation pathways were most prominently regulated (Figure 7d).

### 3.8. Identification of Key Genes in the PPI (Protein–Protein Interaction) Network

A protein–protein interaction (PPI) network of DEGs was constructed using STRING and visualized in Cytoscape. On day 2, 668 nodes and 1158 edges were mapped (Appendix A). After removing single nodes and nodes of degrees below 10, 43 nodes and 287 edges remained (Figure 8a). On day 12, 1025 nodes and 3855 edges were mapped (Appendix A), with 53 nodes and 389 edges after removing single nodes and nodes of degrees below 8 (Figure 8b). The results in the figure only show significantly upregulated differentially expressed genes (*p* < 0.05).

Hub genes were identified using CytoHubba, with the top ten hub genes selected using eight topological methods [45]. On day 2, genes BDD58111.1 and BDD54790.1 were identified by all methods, while BDD56929.1 and BDD61691.1 were identified by seven methods (Appendix A). On day 12, BDD58146.1 was identified by all methods, and BDD58727.1 was identified by seven methods (Appendix A).

## 4. Discussion

In this study, we found that GB significantly influences the polyketide secondary metabolism of *M. purpureus*. The experimental results (Figure 1a–c) demonstrated that the addition of 1 mmol/L GB to the fermentation medium had the most pronounced effect in enhancing the yield of MPs. Specifically, GB exhibited the strongest promotion of red pigment production, reaching a yield of 63.06 U/mL on the 12th day of fermentation and an enhancement of 120.08%. Simultaneously, the same concentration of GB also contributed to an increase in MK yield, with the most significant effect observed on the 12th day of fermentation, achieving a 93.40% increase. The MK yield continued to rise over the fermentation period, peaking on the 15th day (Figure 1d). However, the enhancement effect of GB on MK yield diminished to 49.85% by this time, which was considerably lower than the growth rate observed on the 12th day.

In biological systems, GB functions as a common methyl donor and osmoprotectant. It can act as a metabolic intermediate and participate in microbial biosynthesis and stress protection strategies [20]. For instance, Liu et al. [46] utilized GB as a methyl donor by introducing a GB homocysteine methyl transferase system into *Escherichia coli*, resulting in a 15.9-fold increase in ferulic acid production. Similarly, Kang et al. [47] added GB to the fermentation broth of *Trichosporonoides oedocephalis*, leading to a 50.38% increase in erythritol production. Consistent with these findings, we observed that adding GB to the fermentation broth of *Monascus* enhanced the yield of polyketide metabolites. To our knowledge, this is the first study to explore the impact of introducing a methyl donor into *Monascus* species, providing a scientific foundation for future research and development of *Monascus* polyketide metabolites.

In subsequent research focused on analyzing the effect of GB on the expression of genes related to polyketide synthesis in *Monascus* using RT-qPCR technology, we examined the relative expression levels of 21 genes involved in the regulation of MPs and MK synthesis. The *Monascus* MK biosynthetic gene cluster includes nine genes (*mokA*-*mokI*), among which *mokA*, *mokD*, *mokG*, *mokF*, *mokH*, and *mokI* were significantly upregulated. Based on the RT-qPCR analysis of genes related to MK synthesis, this study summarized the biosynthetic pathway of MK in previous studies and incorporated the gene regulation information into the figure (Figure 9). *MokA* encodes polyketide synthase (PKS), and *mokE* encodes dehydrogenase; these two enzymes synergistically condense acetyl-CoA and malonyl-CoA into dihydromonacolin L, with PKS serving as the key enzyme in the MK synthesis pathway [8]. Notably, overexpression of the *mokE* gene has been reported to result in a 2.5-fold increase in MK production compared to the wild type [48]. Additionally, *Monascus* strains lacking the *mokA* gene do not produce MK [49]. In our experiments, *mokA* and *mokE* genes were upregulated by 1.95 and 4.12 times, respectively, thereby positively contributing to MK synthesis. *mokH* encodes a transcription factor [33], and *mokI* encodes an efflux pump with functions similar to those found in other fungi. Zhang et al. [50] constructed *mokH* gene knockout and overexpression strains, discovering that MK production decreased by 52.05% in the knockout strain and increased by 82% in the overexpression strain. Zhang et al. [36] also overexpressed *mokC*, *mokD*, *mokE*, and *mokI*, resulting in MK production increases of 234.3%, 220.8%, 89.5%, and 10%, respectively. In our experiment, *mokD*, *mokI*, and *mokH* genes were upregulated by 4.40, 3.41, and 6.37 times, respectively, at their peak expression. *mokF* mediates the synthesis of transferase [51], which catalyzes the esterification of 2-methylbutyryl-CoA with the C-8 hydroxyl group of monacolin J, forming the final product, MK. In our study, *mokF* and *mokG* were upregulated by 2.88 and 2.34 times, respectively. However, the expression of *mokB* and *mokC* genes was downregulated; these genes are involved in the oxidation of monacolin L and the synthesis of 2-methylbutyryl-CoA. Chen et al. [8] found that *mokB* deletion mutants lost the ability to produce MK but accumulated the intermediate monacolin J, indicating that *mokB* is responsible for synthesizing the dione side chain of MK. Despite the downregulation of *mokB* and *mokC* expression in our experiment, the upregulation of other genes to varying degrees may account for the slight increase in MK yield.

Furthermore, numerous studies have shown that methionine can directly participate in the synthesis of MK in *Monascus* [20,57,58]. The appropriate addition of methionine to the culture medium can significantly increase MK yield [59]. As a methyl donor, GB can directly participate in the microbial methionine cycle, thereby promoting methionine synthesis [60,61]. This may be one of the reasons why GB, as a methyl donor, can enhance MK yield.

Based on the RT-qPCR analysis of genes related to MPs synthesis, this study integrated findings from previous research to summarize the MPs biosynthetic pathway and included gene regulation data in the illustration (Figure 10). MPs synthesis follows two main branches. The widely accepted pathway suggests that the orange pigments rubropunctatin and monascorubrin are produced through the esterification of a β-ketoacid (from the fatty acid synthase (FAS) pathway) with the chromophore (from the polyketide synthase (PKS) pathway) [62]. The genes *mpFasA2* and *mpFasB2*, homologous to *mpigJ* and *mpigK*, encode the FAS subunit alpha, which is responsible for condensing acetyl-CoA and malonyl-CoA to form the fatty acyl chain [62]. Studies by [63] demonstrated that strains of *M. purpureus* lacking *mpFasB2* accumulated large amounts of chromophore compounds, indicating that a specialized FAS is crucial for modifying azaphilone during pigment biosynthesis. The *mppB* gene, homologous to *mrpigD*, encodes acetyltransferase, which attaches the fatty acyl chain to the azaphilone chromophore [64]. Further research [62] showed that knockout strains lacking *mrpigD*, *mrpigJ*, or *mrpigK* accumulated high levels of polyketide chromophores. These findings highlight the critical roles of *mppB*, *mpFasA2*, and *mpFasB2* in MPs synthesis. In our experiments, these genes were upregulated by 6.50, 1.87, and 2.94 times, respectively.

The *mpPKS5* gene encodes a PKS that condenses acetyl-CoA and malonyl-CoA to produce intermediates in MPs biosynthesis. Balakrishnan et al. [65] found that *M. purpureus* strains lacking *mpPKS5* were unable to synthesize MPs-related compounds. Additionally, *mppF* (orthologous to *mrpigN*) encodes an FAD-dependent monooxygenase involved in forming the pyran ring during MPs synthesis [66]. *mppA* (homologous to *mrpigC*) encodes a ketoreductase that reduces the ω-1 carbonyl group to an alcohol, and *mppD* (homologous to *mrpigG*) encodes a serine hydrolase, which assists *mpPKS5* in pigment production [62]. *mppC* (homologous to *mrpigE*) is an NAD(P)H-dependent oxidoreductase [67]. The products from *mppF*, *mppA*, and *mppC* mutant strains showed that the *mppA*-mediated reduction in the ω-2 ketone is crucial for forming the *Monascus* pigment pyranquinone core, while *mppC* determines the regioselectivity of the Knoevenagel condensation [68]. Our experiments demonstrated that *mpPKS5*, *mppA*, *mppC*, and *mppD* were upregulated by 3.34, 2.08, 3.88, and 4.97 times, respectively. Furthermore, *mpp7* (homologous to *mrpigM*) encodes an acyltransferase involved in dehydrating the ω-1 alcohol, and knockout studies in *M. pilosus* showed that *mpp7* is crucial for regioselective Knoevenagel condensation [69]. *mppE* (homologous to *mrpigH*) encodes an enoyl reductase, and the deletion of *mppE* in *M. purpureus* reduced the production of monascin and ankaflavin, while overexpression increased their levels [70]. Additionally, *mppG* (homologous to *mrpigF*) encodes an oxidoreductase, and inactivation of *mppG* resulted in a marked reduction in orange pigments, but no change in yellow pigments [71]. The transcription factors *mppR1* and *mppR2* play regulatory roles in MPs synthesis, with *mppR2* upregulated 2.09 times in our study, while *mppR1* remained unchanged. Overall, GB addition positively regulated MPs synthesis.

GB can be metabolized into compounds such as choline, trimethylamine, dimethylglycine, creatine, serine, and glycine, which contribute to various microbial synthesis and metabolic pathways [20]. One proposed mechanism for GB’s effect on polyketide metabolism in *Monascus* involves its role as a methyl donor in the methionine cycle. GB loses methyl groups during decomposition, which then participate in the methionine cycle to facilitate the synthesis of methionine and S-adenosylmethionine (SAM) [60,61]. Methionine directly contributes to the synthesis of MK, and its addition to the culture medium significantly enhances MK production [20,57,58,59]. SAM can also directly participate in MPs synthesis [62], and its addition increases MP yield [19]. Therefore, GB likely enhances polyketide production by acting as a methyl donor. From a chemical synthesis perspective, the Knoevenagel condensation, a key step in MPs synthesis, can be catalyzed by quaternary ammonium compounds, which may also increase MP yields [72]. Moreover, GB serves as an osmoprotectant, with its cavity structure binding more water molecules, facilitating substrate transport and enhancing product yields [73].

**Figure 10 jof-11-00273-f010:**
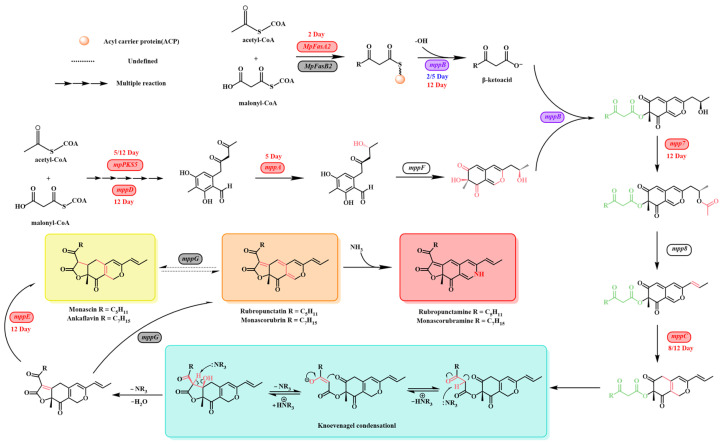
The proposed MPs biosynthetic pathway in *M. purpureus* M1 incorporates the regulation of key genes based on various treatments [56,62,67,70,74,75]. Red indicates upregulation, blue downregulation, and purple indicates mixed regulation at different GB treatment time points, with gray showing no significant regulation and white indicating genes not analyzed.

Additionally, the central regulatory cascade involving *brlA* and *wetA* plays a pivotal role in conidiation. The disruption of *brlA* inhibits conidia formation [76,77], while *wetA* mediates conidia maturation and impermeability [78]. *LaeA*, a global regulator, controls secondary metabolite biosynthesis, growth, and morphogenesis [79]. Overexpression of *laeA* increases secondary metabolite production in *M. purpureus* [40]. SEM revealed significant alterations in spore and hyphal morphology, along with increased extracellular secretion (Figure 4a). RT-qPCR showed that GB treatment upregulated the relative expression of *brlA*, *laeA*, and *wetA* genes (Figure 4b). These results demonstrate that GB enhances fungal growth and development, promoting polyketide metabolite production.

Functional enrichment analysis of differentially expressed genes (DEGs) from RNA-seq data identified upregulated DEGs enriched in processes such as glycolysis, ion transport, and oxidoreductase activity on day 2 (Figure 7a). By day 12 (Figure 7b), upregulated DEGs were involved in protein folding, translation, cytochrome-c oxidase activity, and superoxide dismutase activity. These results, along with GO and KEGG analyses, indicate that GB stimulates the expression of metabolism-related genes during the early growth stages, accelerating polyketide synthesis. In the later growth stages, ribosomal proteins and oxidative phosphorylation genes were activated, facilitating product export and extracellular secretion.

Heat shock proteins (HSPs) HSP10, HSP70, HSP88, and HSP90 were upregulated (Appendix A). These stress-responsive proteins play critical roles in maintaining cellular thermotolerance and protein folding, enhancing metabolic efficiency. HSP70, in particular, aids in polypeptide translocation across membranes [80,81], further supporting GB’s dual role as a methyl donor and osmoprotectant, enhancing polyketide synthesis through different mechanisms.

Lastly, protein–protein interaction (PPI) network analysis identified several hub genes during GB treatment, including ribosomal proteins RPS15, RPS14, and RPS5, which play key roles in stabilizing ribosomal RNA for protein synthesis. Other hub genes, including NDK1, EGD2, and ATP9, may regulate both polyketide biosynthesis and strain growth [82,83]. These results highlight the central role of ribosomal proteins and conserved regulatory genes in the GB-mediated enhancement of MPs synthesis.

## 5. Conclusions

This study offers critical insights into the mechanisms by which GB promotes the growth of *M. purpureus* and enhances the production of key polyketide metabolites, such as MK and MPs. Through RNA-seq analysis, differential gene expression, and protein–protein interaction network modeling, we uncovered GB’s dual role as a methyl donor and osmoprotectant, effectively regulating vital metabolic pathways. GB significantly upregulated genes associated with stress resistance, secondary metabolism, and polyketide biosynthesis, leading to the enhanced production of MK and MPs. Additionally, GB influenced primary metabolism, promoting glycolysis and amino acid metabolism, which enhanced energy production and provided substrates for biosynthetic processes. SEM observations further revealed structural changes in spores and mycelia, confirming GB’s growth-promoting effects. Overall, this study underscores GB’s profound regulatory impact on both primary and secondary metabolism, offering new perspectives on the use of exogenous substances to optimize polyketide production. These findings lay the groundwork for future research on microbial secondary metabolite yield optimization and deepen our understanding of the molecular mechanisms regulated by GB.

## Figures and Tables

**Figure 1 jof-11-00273-f001:**
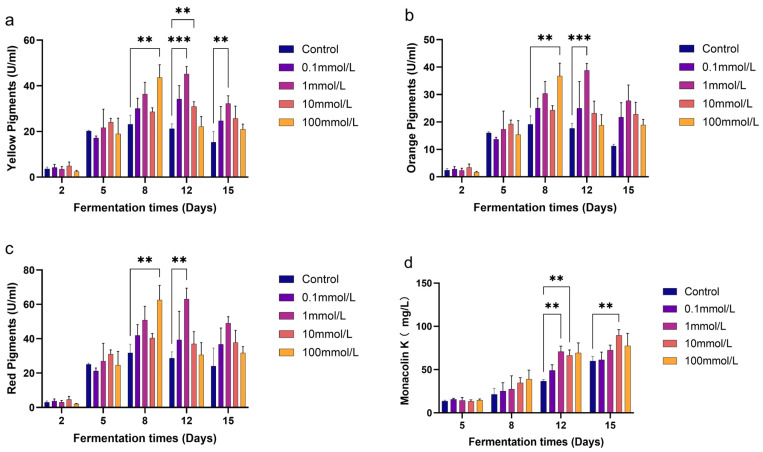
Effect of GB on MPs and MK production of *M. purpureus* M1. (**a**) Red pigment, (**b**) orange pigment, (**c**) red pigment, and (**d**) MK. Data are expressed as the mean ± SD (*n* = 3). ** *p* < 0.01 and *** *p* < 0.001.

**Figure 2 jof-11-00273-f002:**
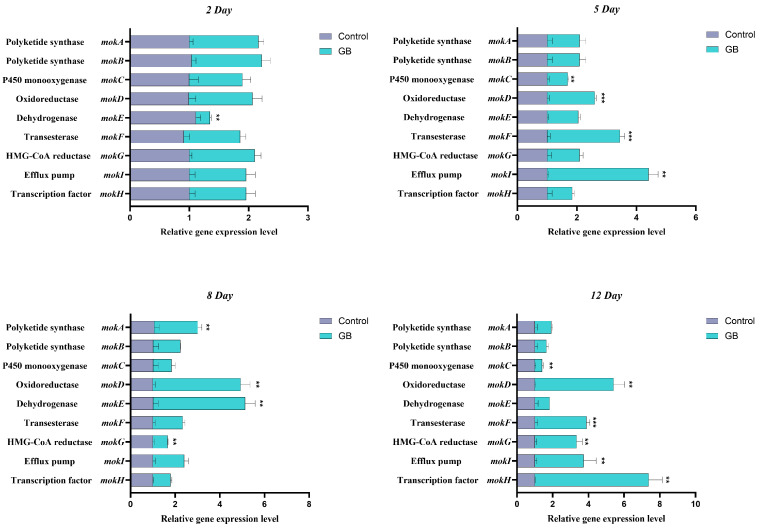
Effect of GB on the relative gene expression levels of MK biosynthesis genes. The *y*-axis label represents gene-related regulatory proteins. Data are expressed as the mean ± SD (*n* = 3). ** *p* < 0.05, *** *p* < 0.001, compared to the control.

**Figure 3 jof-11-00273-f003:**
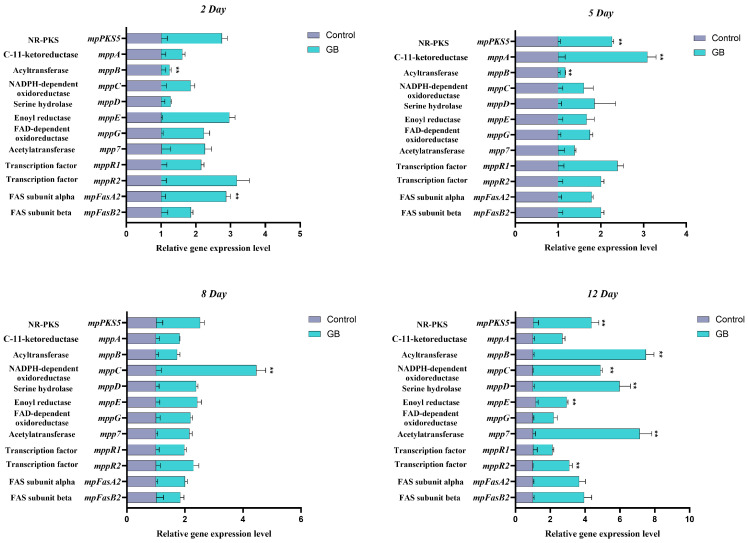
Effect of GB on the relative gene expression levels of MPs biosynthesis genes. The *y*-axis label represents gene-related regulatory proteins. Data are expressed as the mean ± SD (*n* = 3). ** *p* < 0.05, compared to the control.

**Figure 4 jof-11-00273-f004:**
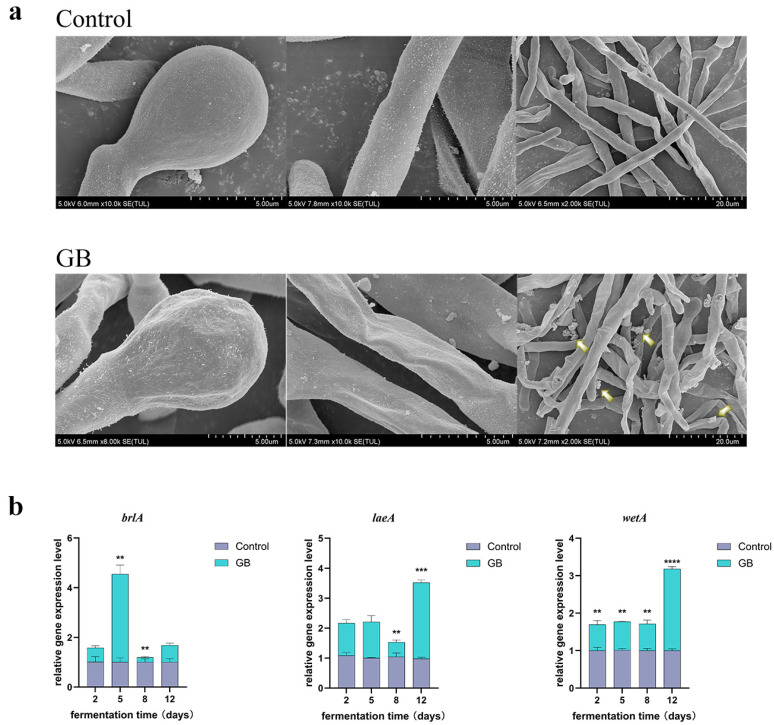
Effect of GB on micromorphology and the expression of asexual development gene of the *M. purpureus* M1. (**a**) Scanning electron microscope images of the mycelium, spore, and extracellular secretions at 2000 ×, 8000 ×, and 10,000 × compared to the control. The yellow arrow points to extracellular secretions. (**b**) Relative expression levels of *laeA*, *brlA*, and *wetA* based on RT-qPCR. Data are expressed as the mean ± SD (*n* = 3). ** *p* < 0.05 and *** *p* < 0.001, **** *p* < 0.0001, compared to the control.

**Figure 5 jof-11-00273-f005:**
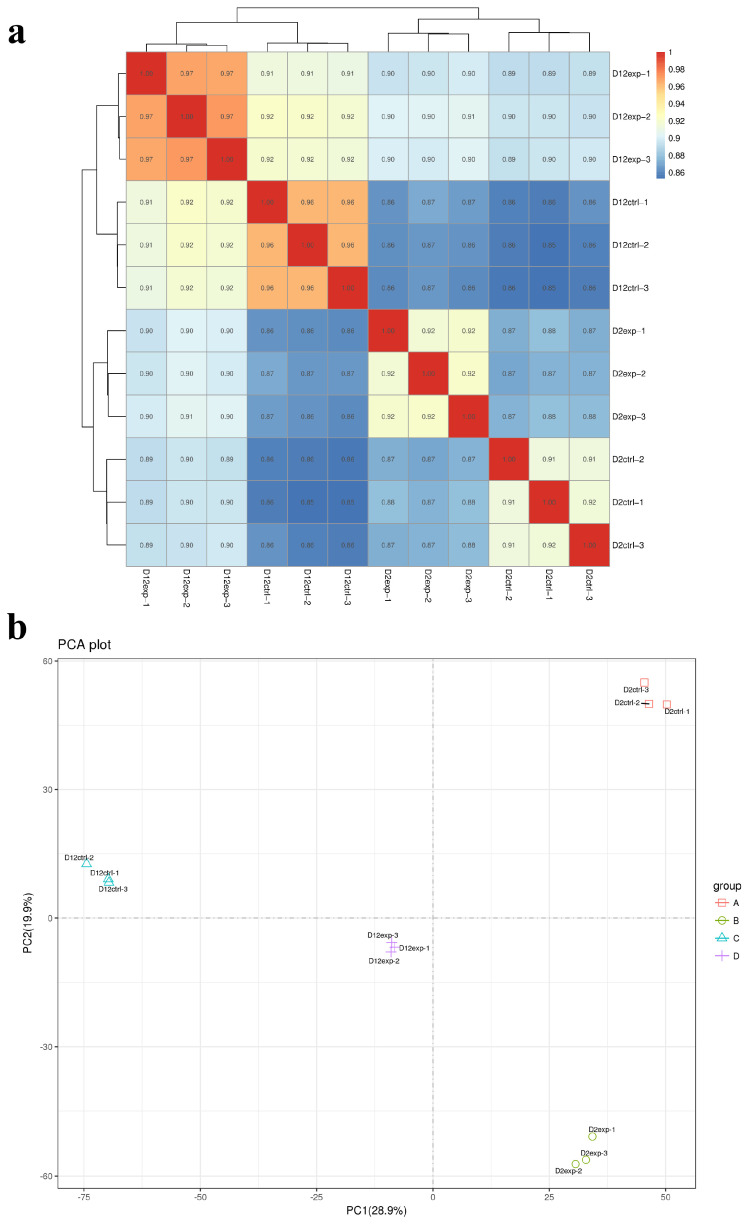
Pearson correlation analysis and PCA. (**a**) Pearson correlation analysis between all RNA samples. A redder color indicates a stronger correlation. (**b**) PCA plot: Group A (control on day 2), Group B (GB-treated on day 2), Group C (control on day 12), Group D (GB-treated on day 12).

**Figure 6 jof-11-00273-f006:**
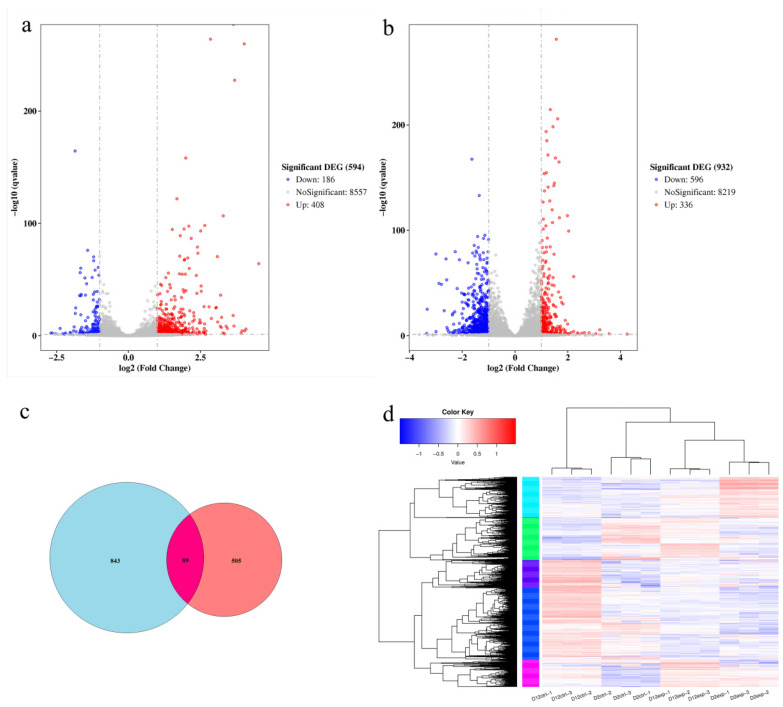
Analysis of differential gene expression. (**a**) Volcano plot of DEGs on day 2 of fermentation. Red dots represent upregulated genes, blue dots represent downregulated genes. (**b**) Volcano plot of DEGs on day 12 of fermentation. (**c**) Venn diagram showing shared DEGs between day 2 and day 12. (**d**) Hierarchical clustering of DEGs based on FPKM values. Red indicates highly expressed genes; blue indicates lowly expressed genes.

**Figure 7 jof-11-00273-f007:**
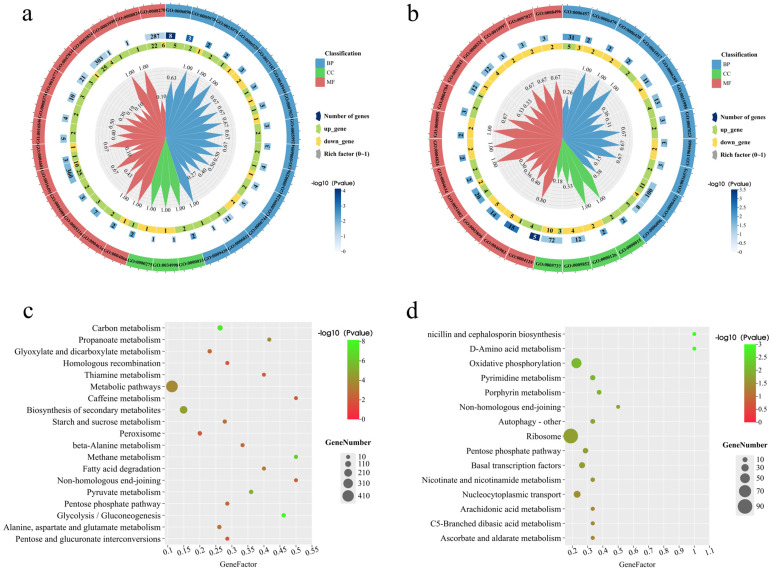
Gene expressions and GO functional analysis of DEGs regulated by GB. (**a**,**b**) GO functional annotation of DEGs on day 2 and day 12 of GB treatment. (**c**,**d**) KEGG enrichment analysis of DEGs on day 2 and day 12 of GB treatment.

**Figure 8 jof-11-00273-f008:**
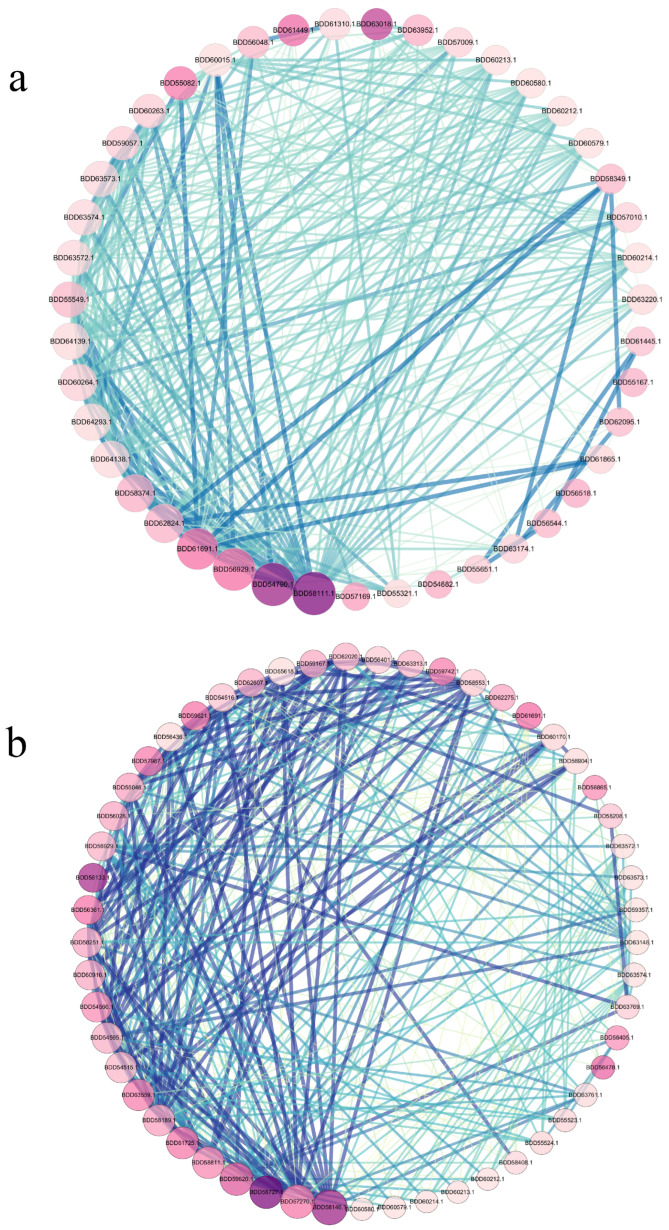
The protein–protein interaction (PPl) network was constructed by STRlNG and Cytoscape. (**a**,**b**) represent the PPI networks for day 2 and day 12 of GB treatment, respectively. The larger nodes represent higher node connect degree. Darker node colors represent higher betweenness centrality. Edges that are darker and thicker indicate a higher combined score. Higher centrality of a node indicates greater importance within the network, and a higher combined score reflects stronger protein–protein interaction.

**Figure 9 jof-11-00273-f009:**
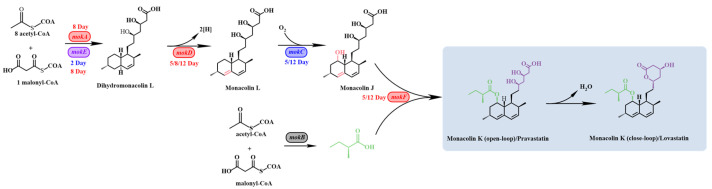
Proposed biosynthetic pathway for MK in *M. purpureus* M1 [52,53,54,55,56]. Red represents upregulation, blue represents downregulation, and purple represents upregulation and downregulation in different GB treatment days.

## Data Availability

The data that support the findings of this study are available on request from the corresponding author. The data are not publicly available due to privacy or ethical restrictions.

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
