# Peer review of "RNA-Seq Insight into the Impact and Mechanisms of Methyl Donor and Glycine Betaine Osmoprotectant on Polyketide Secondary Metabolism in Monascus purpureus M1"

_jof, 2025, doi:10.3390/jof11040273_

Round 1

Reviewer 1 Report

Authors investigated the effects of GB on the production of pigments and MK produced by Monascus purpureus M1, and analyzed the reasons that addition of GB can increase their yields from the gene transcription level, broadening the ways to increase the production of pigments and MK. However, the yield of secondary metabolites is related to the expression of genes in their biosynthetic pathway and is also subject to many regulations at different level. Although the author analyzed the gene expression from the transcriptome, there are few genes discussed on the  regulatory genes. So more detailed analysis of the data should be carried out. In addition, since GB is related to stress resistance, it would be better to perform some stress resistance experiments.

Line 88-90: please describe the  species information.

Line 535-538:The sequence of fig a and fig b should be changed.

Author Response

Dear Editor and Reviewers,

Thanks for your great concern and comments on my manuscript. We have addressed the reviewers’ concerns in detail. A point-by-point response to the comments made by the reviewers has been provided. The changes addressing reviewers’ comments were listed below. All issues were carefully addressed in the response to reviewer and implemented in the revised manuscript. The changes were highlighted in yellow in Revised Manuscript.

Reviewer #1:

Comments 1: Authors investigated the effects of GB on the production of pigments and MK produced by Monascus purpureus M1, and analyzed the reasons that addition of GB can increase their yields from the gene transcription level, broadening the ways to increase the production of pigments and MK. However, the yield of secondary metabolites is related to the expression of genes in their biosynthetic pathway and is also subject to many regulations at different level. Although the author analyzed the gene expression from the transcriptome, there are few genes discussed on the  regulatory genes. So more detailed analysis of the data should be carried out. In addition, since GB is related to stress resistance, it would be better to perform some stress resistance experiments.

Response 1: Thank you for your valuable suggestion. In our previous studies, we have identified the regulatory genes involved in the biosynthesis of Monascus secondary metabolites, including Monascus pigments (MPs) and Monacolin K (MK) (DOI: 10.3390/jof8020179; DOI: 10.1016/j.lwt.2022.114225). Therefore, we specifically focused our RT-qPCR analysis on these regulatory genes. The subsequent data indicated that these regulatory genes were significantly upregulated to varying degrees upon betaine treatment (Fig. 2; Fig. 3). Our subsequent transcriptomic analysis of differentially expressed genes in this strain aimed to comprehensively investigate the regulatory effects of betaine from multiple perspectives.

Meanwhile, we believe that your suggestion to include additional stress resistance experiments is very reasonable. This paper primarily focuses on the effects and mechanisms of betaine on secondary metabolites. Additionally, in the discussion section, we have mentioned the upregulation of some stress-responsive related genes. Stress resistance experiments represent a more innovative aspect, and we plan to conduct further experiments and perform a more in-depth analysis in the future.

Comments 2:Line 88-90:please describe the species information.

Response 2:Thank you very much for your suggestions. We have added information regarding microbial species. Additionally, after careful verification, we found that betaine is widely present in both animals and plants. Here are some relevant references (DOI: 10.1007/s11240-024-02832-3; DOI: 10.1093/ajcn/80.3.539; DOI: 10.1016/s1095-6433(01)00410-x). Consequently, we have decided to revise "animals" as "vertebrates" and revise "plants" to Beta vulgaris ssp., Bougainvillea spp., and others. Furthermore, we have also updated the relevant references accordingly.

Comments 3:Line 535-538:The sequence of fig a and fig b should be changed.

Response 3: Thank you for pointing out this issue in the manuscript. We have exchanged the order of Fig. a and Fig. b.

Reviewer 2 Report

The study is interesting. However, I do not consider the pigments analysis appropriate and the main focus of the study is on pigments production. Have you analyzed the pigments produced by the used Monascus strain in any case? Are you sure that the pigments profile is not changing with the change of the culture conditions?

Is the strain, which you used in the study, really Monascus purpureus species? Given the recent insight into the Monascus taxonomy (see https://www.frontiersin.org/journals/microbiology/articles/10.3389/fmicb.2023.1199144/full) it seems to be Monascus pilosus (ruber). Here again the pigments profile would be helpful. 

What is the nature of the extracellular secretions which can be seen in the Fig.4?

The predicted production times of the various regulators of pigment production (Fig. 10 in the discussion) do not correspond to Fig. 1, which shows an increase in pigment production with fermentation time up to day 12.

Author Response

Dear Editor and Reviewers,

Thanks for your great concern and comments on my manuscript. We have addressed the reviewers’ concerns in detail. A point-by-point response to the comments made by the reviewers has been provided. The changes addressing reviewers’ comments were listed below. All issues were carefully addressed in the response to reviewer and implemented in the revised manuscript. The changes were highlighted in yellow in Revised Manuscript.

Reviewer #2:

Comments 1:The study is interesting. However, I do not consider the pigments analysis appropriate and the main focus of the study is on pigments production. Have you analyzed the pigments produced by the used Monascus strain in any case? Are you sure that the pigments profile is not changing with the change of the culture conditions?

Is the strain, which you used in the study, really Monascus purpureus species? Given the recent insight into the Monascus taxonomy (see https://www.frontiersin.org/journals/microbiology/articles/10.3389/fmicb.2023.1199144/full) it seems to be Monascus pilosus (ruber). Here again the pigments profile would be helpful.

Response 1: Thank you very much for your question. We apologize for not having analyzed the pigments produced by used Monascus strains in this study. This could indeed be an interesting research direction for future work. Additionally, we believe that the pigments profile may undergo slight variations depending on the culture conditions. However, due to the presence of a control group, such variations are negligible and do not affect the overall experimental outcomes. Furthermore, we did not find any relevant studies on the pigments profile of Monascus purpureus M1. This indicates that study also holds a certain level of novelty, and we may explore this aspect in future research.

Furthermore, after carefully reviewing the article you provided, we had an in-depth discussion. The study advocates identifying M. pilosus, M. ruber, and M. purpureus at the genomic level. We regret that we are unable to answer your question based on the provided reference, because we have not conducted whole-genome sequencing of Monascus purpureus M1. However, the strain we used is a laboratory-preserved strain, and it has been utilized in multiple previously published studies. Here are some relevant references (DOI: 10.3390/jof8020179; DOI: 10.1016/j.lwt.2022.114225; DOI: 10.1007/s00253-020-10379-4; DOI: 10.1002/yea.3831; DOI: 10.1002/jsfa.11218; DOI: 10.1016/j.funbio.2020.03.010; DOI: 10.3389/fmicb.2020.610471). Based on this extensive research, we are confident that this strain indeed belongs to the Monascus purpureus species.

Comments 2:What is the nature of the extracellular secretions which can be seen in the Fig.4?

Response 2:Thank you for your question. We have reviewed previous studies, we did not find specific descriptions of the properties of extracellular secretions. However, we discovered that the upregulation of the global regulator laeA may mediate the production of extracellular secretions, which is consistent with our findings. Here are some relevant references (DOI: 10.1007/s00253-020-10379-4; DOI: 10.1016/j.lwt.2022.114225). Investigating the properties of extracellular secretions is indeed an interesting research direction and we may explore this aspect further in future research.

Comments 3:The predicted production times of the various regulators of pigment production (Fig. 10 in the discussion) do not correspond to Fig. 1, which shows an increase in pigment production with fermentation time up to day 12.

Response 3:Thank you very much for your question. This is an excellent point, and we also took note of this issue during data analysis. First, the gene regulation timing mentioned in Fig. 10 is based on the RT-qPCR data presented in Fig. 3. Previous research has shown (DOI: 10.3390/jof8020179) that there is no clear correlation between the timing of Mps biosynthetic gene regulation and the peak pigment yield. Second, as we all know, the MPs biosynthetic pathway is highly complex, involving more than a dozen biotransformation reactions. To date, there are still several unresolved controversies regarding MPs biosynthesis (DOI: 10.3389/fmicb.2022.951266), such as the expression and regulation of the PKS gene cluster. Therefore, it is reasonable that the gene regulation timing does not strictly align with the peak MPs yield. Furthermore, betaine, as a methyl donor and osmoprotectant, can influence multiple metabolic pathways. This diversity in regulatory mechanisms means that the pathways leading to increased secondary metabolite production are varied, making the relationship between gene regulation timing and MPs yield more complex and less predictable.

Reviewer 3 Report

The manuscript is devoted to the study  the effects of glycine-betaine on polyketide secondary metabolism in Monascus purpureus.

The authors have done a great job.

The manuscript may be published in J Fungi.

There are minor comments that would be desirable to correct.

L 40

 This organism -  This microorganism

L 147  L 151 – 5%, 10 %  inoculation  rate - what is meant by this?

Figure 1 a) yellow

RT-qPCR Analysis of MK Biosynthesis Related Genes and RT-qPCR Analysis of MPs Biosynthesis Related Genes – what concentrations of GB were used in the study?

It would be interesting to know the effect of GB on biomass accumulation of fungus.

The manuscript is devoted to the study  the effects of glycine-betaine on polyketide secondary metabolism in Monascus purpureus.

The authors have done a great job.

The manuscript may be published in J Fungi.

There are minor comments that would be desirable to correct.

L 40

 This organism -  This microorganism

L 147  L 151 – 5%, 10 %  inoculation  rate - what is meant by this?

Figure 1 a) yellow

RT-qPCR Analysis of MK Biosynthesis Related Genes and RT-qPCR Analysis of MPs Biosynthesis Related Genes – what concentrations of GB were used in the study?

It would be interesting to know the effect of GB on biomass accumulation of fungus.

Author Response

Dear Editor and Reviewers,

Thanks for your great concern and comments on my manuscript. We have addressed the reviewers’ concerns in detail. A point-by-point response to the comments made by the reviewers has been provided. The changes addressing reviewers’ comments were listed below. All issues were carefully addressed in the response to reviewer and implemented in the revised manuscript. The changes were highlighted in yellow in Revised Manuscript.

Reviewer #3:

Comments 1:L 40  This organism -  This microorganism

Response 1:Thank you very much for your correction. We have revised the word accordingly.

Comments 2:L 147  L 151 – 5%, 10 %  inoculation  rate - what is meant by this?

Response 2: Thank you very much for your suggestion. We recognized an issue with our wording in this section and have revised it to ensure a more scientifically accurate description.

Comments 3: Figure 1 a) yellow

Response 3:Thank you very much for your correction. We have revised the incorrect word in Fig. 1a.

Comments 4:RT-qPCR Analysis of MK Biosynthesis Related Genes and RT-qPCR Analysis of MPs Biosynthesis Related Genes – what concentrations of GB were used in the study?

Response 4:Thank you for your question. We have added the relevant information in Lines 198–200.

Round 2

Reviewer 1 Report

The writing format of all microorganisms in this article is incorrect, especially in the part of the references. Please carefully check and make corrections. 

Line 539-540:The Figures should be arranged in alphabetical order.

Author Response

Dear Editor and Reviewers,

Thank you for your attention to our manuscript and your valuable comments. We have thoroughly revised the manuscript in full accordance with the reviewer raised in the second round of review. The changes addressing reviewers’ comments were listed below. All issues were carefully addressed in the response to reviewer and implemented in the revised manuscript. The changes were highlighted in yellow in Revised Manuscript.

Reviewer #1:

Comments 1: The writing format of all microorganisms in this article is incorrect, especially in the part of the references. Please carefully check and make corrections.

Response 1:Thank you for your correction. We sincerely apologize for this oversight during the revision of the references. The formatting has now been corrected as required.

Comments 2:Line 539-540:The Figures should be arranged in alphabetical order.

Response 2:Thank you for your suggestion. We have corrected the issue you pointed out.

Reviewer 2 Report

I have no other comments. 

No comments.

Author Response

Thank you very much for your attention to our manuscript and your valuable suggestions. We extend our sincere regards.